nanotechnology

gold nanostars, HEPES, polyvinylpyrrolidone, seedless, spectroscopy

**Author for correspondence:**
Danielle Wingrove Mulder
e-mail: danitedder@gmail.com

# Modified HEPES one-pot synthetic strategy for gold nanostars

Danielle Wingrove Mulder[1], Masauso Moses Phiri[1], Anine Jordaan[2] and Barend Christiaan Vorster[1]

[1]Center for Human Metabolomics, North-West University, Hoffman street, Potchefstroom, South Africa
[2]Chemical Resource Beneficiation (CRB), North-West University, Potchefstroom, South Africa

DWM, 0000-0002-6970-7392; MMP, 0000-0001-7653-7988; BCV, 0000-0003-2371-288X

Gold nanostars are being used more regularly in the biosensing field. Despite their useful attributes, there is still a need to optimize aspects of the synthesis and stability. The seedless, synthetic method comprising 4-(2-hydroxyethyl)-1-piperazineethanesulfonic acid (HEPES) is a facile, rapid method; however, it produces heteromorphic nanostars. The modification of a HEPES method resulted in a silver-assisted, seedless gold nanostar synthesis method. The nanostars resulting from this method were monodispersed, multi-branched and approximately $37 \pm 2$ nm in diameter. It proved to be a repeatable method that produced homogeneous and robust nanostars. Once functionalized with polyvinylpyrrolidone 10 000, the new nanostars were observed to be stable in various environments such as salt, ionic strength and cell culture medium. In conclusion, the addition of the silver nitrate improved the morphology of the reported HEPES nanostars for the purpose of nanobiosensor development.

## 1. Introduction

Gold nanoparticles have been a favoured option in biosensing as a result of their facile synthetic methods, catalysis, biocompatibility, large surface area, and optical and thermal properties [1]. Gold nanostars are debatably one of the most promising morphologies, as the protruding arms add to the plasmonic contributions and the lightning rod effect. This enhances the electromagnetic field, and thus they have the highest enhancement factors in indirect and direct localized surface plasmon resonance (LSPR) biosensing [2]. The surface plasmon resonance can be altered by controlling the arm density and length of the nanostar architecture without altering the overall dimensions [3]. In 2012 Stevens and colleagues

showed that LSPR shifts in response to a biorecognition event. By using the hydrogen peroxide generated by glucose oxidase, the silver ions were reduced around the gold nanostar nanosensors producing this shift [4].

Despite the versatility of gold nanostars, there are still factors requiring attention regarding synthesis. Synthesis methods with control that ensures reproducible monodisperse stars, with sufficient yield of the morphology of interest, well-defined properties and potential scale-up are currently still lacking. Due to such limitations, there has been a substantial decrease in nanostar interest and research [5–7]. Experimental and theoretical studies are still needed to understand the fundamental behaviour of these nanoparticles [7]. Not only are morphological aspects a challenge, but reagent toxicity is also a concern. Environmentally and biologically hazardous reagents such as dimethyl formamide and sodium borohydride are used in most documented methods. Interest has thus been ignited to find alternative 'green' synthesis methods such as using Good's buffers [8,9].

Two different synthesis categories exist for gold nanostars, namely, seeded and seedless methods. The seeded nanostar synthesis uses gold nanoparticle seeds that are grown and guided to produce branches. These methods require multiple steps which may add considerably to batch variations. The post-synthesis purification is complicated by use of surfactants [5]. The seedless method, on the other hand, uses fewer steps and reagents. The methods are mostly simple and have less post-synthesis purification complications. However, they mostly yield highly polydispersed samples due to insufficient precision of the reaction parameters, such as reagent concentrations, pH and temperature [5,10].

2-[4-(2-hydroxyethyl)piperazinyl]ethanesulfonic acid (HEPES), one of the Good's buffers used in seedless methods, has been found to produce more branched nanoparticles, less post-synthesis purification complications, higher particle stability and good potential for scalability [11–13]. HEPES is a zwitter-ionic organic buffering agent that has minimal salt and temperature effects and has high solubility in water. It is used in cell culture because of its low permeability to cell membranes [10]. HEPES has predominantly been used among the Good's buffers in nanostar synthesis as a precise shape-directing agent [14]. The piperazine moiety in HEPES generates nitrogen-centred free radicals that reduce the gold ions ($AuCl_4^-$), rendering HEPES as both a good weak reducing and capping agent [13,15,16]. Since the reported use of HEPES for colloidal gold nanostars synthesis by Xie *et al.*, it has been used in many other seedless methods with varied modifications [13]. Saverot and co-workers recently reported a two-step approach using HEPES and the addition of $HAuCl_4$ for gold nanostar growth and branch length [10]. However, fine tuning of the optical properties of gold nanostars by controlling the morphology uniformity has not been reported in seedless methods [14].

In this study, we propose a simple, one-pot, silver-assisted green synthesis method of seedless gold nanostar using HEPES. Although silver nitrate has been used as a shape directing agent in seeded methods, to our knowledge, it has not been reported in a HEPES-mediated synthesis of gold nanostar [17,18]. As opposed to a two-step method, this method was a one-step synthesis where silver nitrate was used as an additional shape-directing agent to assist HEPES in obtaining more monodispersed multi-branched gold nanostars. The method also used the least toxic reagents. The produced gold nanostars were investigated for stability in environmental conditions mostly applied in bioassays for potential applications for biosensing work.

# 2. Material and methods

## 2.1. Nanostar synthesis

The chemicals used in this study were 4-(2-hydroxyethyl)-1-piperazineethanesulfonic acid (HEPES), gold (III) chloride hydrate ($HAuCl_4.4H_2O$), polyvinylpyrrolidone (PVP) 10 000, tris borate EDTA buffer (TBE), sodium hydroxide, hydrochloric acid and silver nitrate, which were purchased from Sigma-Aldrich. The cell culture medium used was Ham's F-12K (Kaighn's) medium and supplemented with 10% fetal bovine serum, which were purchased from ThermoFisher Scientific.

### 2.1.1. Seedless (−Ag)

The seedless nanostars were synthesized according to Xie *et al.* [13]. Briefly, 3 ml ultrapure water (Millipore, 18.2 MΩ (ddH$_2$O)) was added to 2 ml 100 mM HEPES buffer (pH 7.4), followed by the addition of 20 µl of a 50 mM gold (III) chloride hydrate aqueous solution. After gentle end-to-end inversion mixing the solution was left to stand at room temperature for approximately 30 min at which time the solution turned a greenish blue colour. Then 600 µl 25 mM PVP 10 000 was added to the solution and left to stand overnight at room

temperature after gentle mixing. The solution was then centrifuged at 2170$g$ for 50 min after which the soft pellet was washed and resuspended in 500 µl ddH$_2$O. The stabilization of other coating polymers was not assessed as this was done in another study found in literature using HEPES buffer. What was found was that polyethylene glycol (PEG) was not a good stabilizing agent compared with polyvinylpyrrolidone (PVP) and CTAB [8]. PVP was, thus, chosen as it was the less toxic reagent when compared with CTAB [19,20].

### 2.1.2. Silver-assisted seedless (+Ag)

The silver assisted nanostars were synthesized according to the seedless method as described above, however, with the addition of 1 mM aqueous silver nitrate solution. Directly after the gold (III) chloride hydrate was added 2, 4, 6, 8 and 10 µl silver nitrate solution was added, respectively, followed by gentle inversion. Based on the nanostar morphology, the 4 µl +Ag sample was chosen for the preceding experiments to which PVP was added as described above.

## 2.2. Characterization

The characterizations of the nanostars were done in accordance with the methodologies as suggested by ISO TR13014:2012 [17].

### 2.2.1. Size, morphology and elemental composition

The diameter, morphology and elemental composition of the uncoated nanostars were determined with high-resolution transmission electron microscopy (HR-TEM) and energy-dispersive X-ray spectroscopy (EDS) on a Tecnai F20 high-resolution transmission electron microscope. Sample preparation was done by spotting the nanostars on to a copper grid and allowing them to dry. Particle distribution analysis was done from the TEM images using ImageJ software.

Dynamic light scattering (DLS) was used to estimate the hydrodynamic diameter of the PVP capped nanostars as this would be the size the nanostars would be in solution [21,22]. DLS was performed on a Zetasizer Nano (Malvern) in backscatter mode using Zetasizer version 6.20 software and stoppered polystyrene cuvettes limiting dust contamination.

### 2.2.2. Surface functionalization and charge

Successful PVP capping of the nanostars was demonstrated with nuclear magnetic resonance (NMR) spectral matching of the spectra of washed nanostars to that of a PVP standard. The NMR was performed at 500 MHz on a Bruker Avance III HD NMR spectrometer equipped with a triple-resonance inverse (TXI) 1H (15N, 13C) probe head. Adequacy of capping, as indicated by a negative surface charge and limited aggregation was assessed with agarose gel electrophoresis. The electrophoresis was done using 0.5% agarose and 0.5× tris borate EDTA buffer (TBE buffer) at pH 8. The samples were prepared using 8 µl nanostars mixed with 4 µl 50% glycerol and ran at 40 V for approximately 30 min [23].

### 2.2.3. Spectral properties

Finally, the absorbance spectrum of the nanostars was determined using a HT Synergy (BioTEK) microplate spectrophotometer operated between 400 and 800 nm, and Gen5.1 as the corresponding software.

## 2.3. Environmental effect on nanostar stability

The environmental effect was investigated by exposing the nanostars to various solutions. Solutions of 150 mM NaCl, pH 4–8 as well as supplemented (with fetal bovine serum) and unsupplemented (no fetal bovine serum) cell culture medium were added to the nanostar solutions in a 1 to 1 ratio in a 96-well microplate and mixed by pipetting. These parameters were chosen as most bioassay work would require the nanostars to be stable in these conditions. Each sample was prepared in triplicate and absorbance spectra were obtained after 24 h incubation.

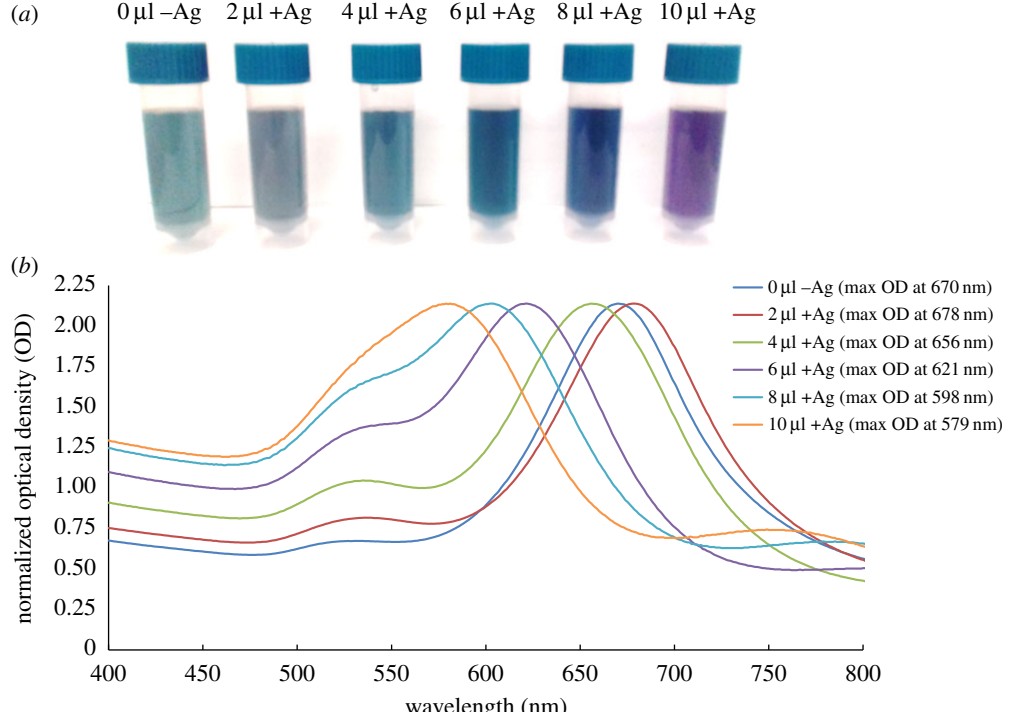

**Figure 1.** Influences of different silver nitrate concentrations to nanostar synthesis. (*a*) Colorimetric changes and (*b*) UV−Vis spectra.

## 3. Results and discussion

When the silver nitrate was added to the synthesis the colour of the solutions varied from a greeny blue (0 µl) to grey (2 µl), blue (4 µl), dark blue (6 µl), blue-purple (8 µl) and purple (10 µl) (figure 1*a*).

The spectra seen in figure 1*b* show a blue shift as the silver nitrate concentration increased (except for the 2 µl which red shifted slightly). The average wavelength at which the maximum optical density occurred over 12 separately synthesized samples of the 4 µl +Ag was found to be 654 nm $\pm$ 5 nm (figure 1*b*) with an average hydrodynamic diameter of 51.36 nm $\pm$ 0.70 nm which was indicative of gold nanostars [24]. The morphological change of the particles presented in the HR-TEM images (figure 2), gave a plausible explanation of the observed spectral shifts in figure 1.

Nanostars synthesized with 2 µl of the silver nitrate solution displayed morphology similar to that of −Ag nanostars, while those synthesized with 6, 8 and 10 µl, respectively, were progressively more spherical in nature (figure 2*a*,*b*,*d*−*f*). The +Ag nanostars synthesized using 4 µl silver were then used for the remaining experiments as they were more star-shaped in morphology (figure 2*c*). The dispersity of these nanostar-shaped particles and their size were further assessed. These results are seen in figure 3.

The particle distribution analysis was done using TEM images of 175 particles. The average +Ag particle diameter was found to be approximately 37 $\pm$ 2 nm. The full width at half maximum (FWHM) for −Ag nanostars was 96 nm, which agreed with Xie *et al.* [13], and for the +Ag nanostars was 72 nm. This showed that the +Ag nanostars were more monodispersed than the −Ag nanostars, as the FWHM for +Ag nanostars was narrower than that obtained for the −Ag nanostars. Further analysis of the nanostar morphology is seen in figure 4.

Analysis of the morphological differences between the −Ag and the +Ag nanostars is presented in figure 4. The −Ag nanostars were heterogeneous in morphology and tended to agglomerate (figure 4*a*). The predominant morphology was the four-armed nanostar (figure 4*b*) which agreed with previous observations [13]. The addition of 4 µl of the silver nitrate solution changed the heterogeneity of the nanostars into a more homogeneous multi-branched morphology by assisting in the anisotropic growth of the nanostar branches (figure 4*c*). The nanostars contained approximately 14 arms with 10 arms protruding around the nanostar and approximately four arms protruding outwards (figure 4*d*). Of the 100 nanostars observed, 65% were the 14-armed nanostars, whereas the remainder were more spherical with a few short protruding arms. The −Ag nanostars were more heteromorphic as they had a range of 1−8 arms (figure 4*a*) which agreed with those found in Xie *et al.* [13]. The

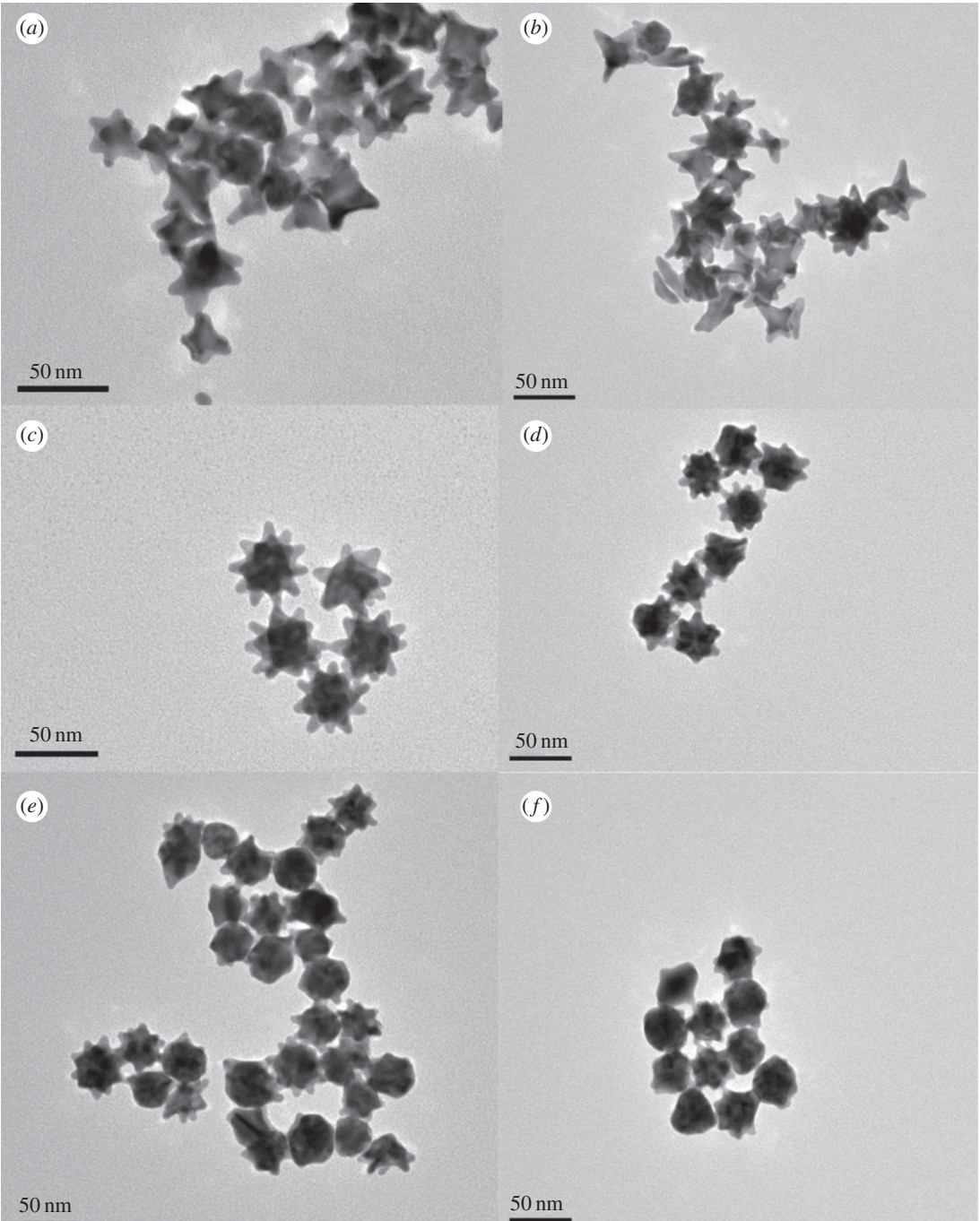

**Figure 2.** Representative HR-TEM images of the nanoparticle morphology. (*a*) 0 μl −Ag, (*b*) 2 μl +Ag, (*c*) 4 μl +Ag, (*d*) 6 μl +Ag, (*e*) 8 μl +Ag, and (*f*) 10 μl +Ag gold nanostars.

presence of the shoulder peak seen in the +Ag spectrum at 525 nm in figure 3, could have been attributed to the presence of the large spherical particles or to the transverse localized surface plasmon resonance depending on the core size of the anisotropic nanostars. The main peak corresponded to the major axis and the shoulder peak to the minor axis tip of the star [25,26].

The size of the +Ag nanostars were smaller than the −Ag ones of 46 nm ± 3 nm. The growth of the crystal lattice was in the ⟨111⟩ direction [27] and had a lattice spacing of 0.233 nm (figure 4*d* insert), which was in the bulk gold range [13]. EDS analysis revealed an elemental composition consisting of gold (79.4%) and copper (20.6%) which was from the copper grids used during analysis. Although elemental silver was not detected by the EDS analysis, the silver did aid the HEPES in disrupting the lattice of the gold nanostars more, resulting in more homogeneous branches as opposed to the −Ag synthesized nanostars.

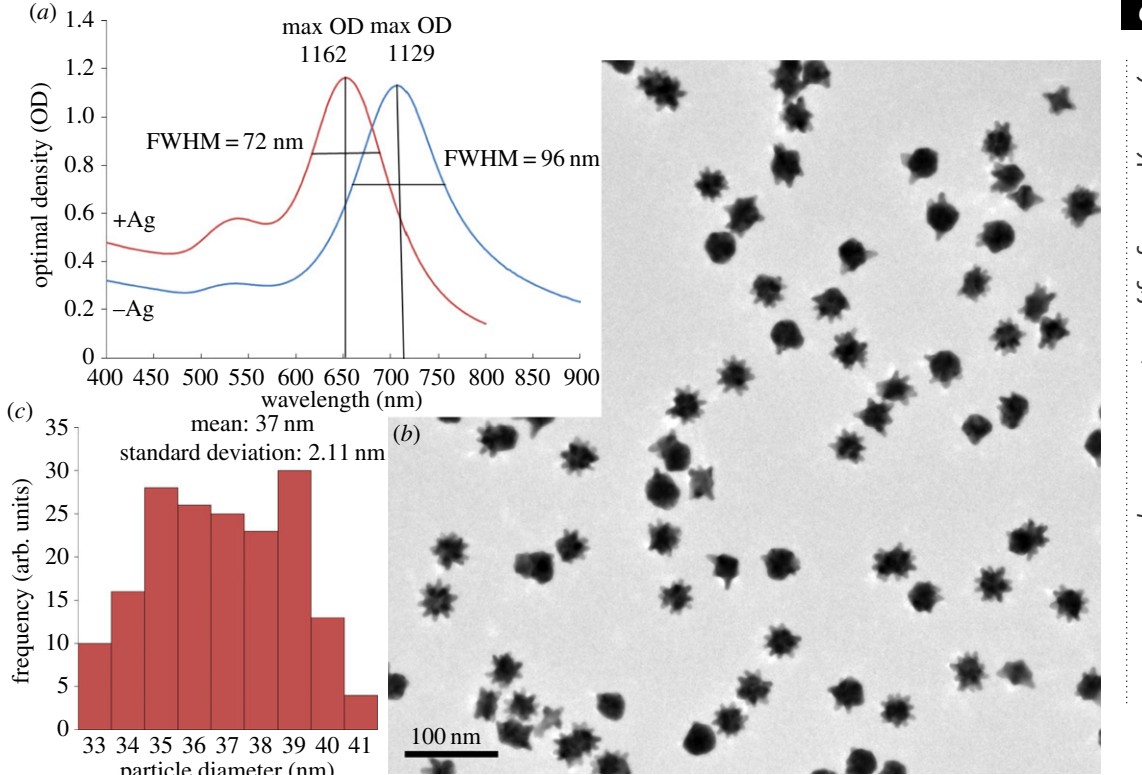

**Figure 3.** Particle size distribution and monodispersity analysis. (a) UV−Vis spectra indicating the full width at half maximum value for both −Ag and +Ag nanostars. (b) TEM image of +Ag nanostars. (c) Particle diameter distribution using TEM images.

These new nanostars were different in morphology and synthesis when compared with other seedless HEPES synthesized nanostars found in literature. Webb *et al.* [3] synthesized particles using a similar method to Xie *et al.* [13] but differed the concentrations and pH of HEPES as well as the HAuCl₄ concentration. The final product produced a limited number and heterogenous multibranched nanostars [3]. In a follow-up article Xie *et al.* synthesized nanostars by changing the HEPES concentration, which led to more homomorphic branched nanostars which had a large core and extremely short arms [28]. Liu *et al.* synthesized nanostars with four long branched arms by manipulating the HEPES concentration and solution temperature [29]. Longer arms and branch density resulted in a more sensitive response to morphological change which was a desirable aspect for a biosensor [3,29]. The particles synthesized by Chen *et al.* were synthesized by adding surfactants CTAB, PVP and PEG to the HEPES during synthesis. The morphology of the particles ranged from spherical, triangular to a three-armed star [8]. Minati *et al.* synthesized nanostars using a seedless approach with hydroxylamine and HAuCl₄; although this was also a very simple method, the nanostars that were formed had a large core with extremely short arms [5]. Araújo-Chaves *et al.* synthesized a complex particle conglomerate by adding phosphate buffered saline to the HEPES which also had a large core and extremely short arms [15]. In comparison, these new synthesized nanostars with the simple addition of AgNO₃ resulted in a gold nanostar that had more homogeneous arms and which was sufficient in length for the use of enzyme and antibody attachment.

Once capped with PVP, both the −Ag and +Ag migrated to the cathode when compared with uncapped +Ag controls during agarose gel electrophoresis. In addition, NMR spectral matching indicated the presence of PVP in the washed samples. Taken together these results indicated sufficient PVP capping of the nanostars which aided in centrifugation and storage stability. As mentioned earlier, HEPES also acts as a capping agent which proved true in this study as the HEPES was still present after multiple washes (figure 5). This was, however, of little concern for the future use of these particles as they will be functionalized with other compounds of interest.

The UV−Vis spectral comparisons of −Ag and +Ag nanostars after 24 h exposure to various environments are presented in figure 6. Both nanostars were not stable in the incomplete medium, however, once the medium was supplemented then the nanostars remained reasonably stable with a slight red shift in the spectrum when compared with the control nanostar solution. The instability

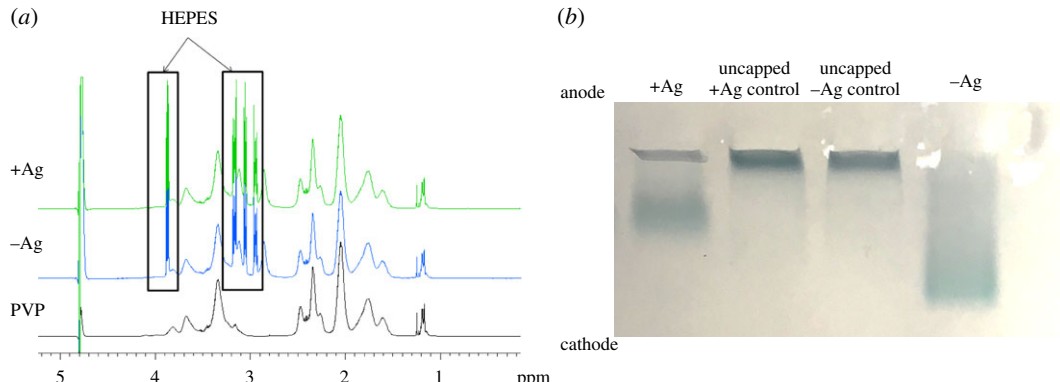

**Figure 4.** HR-TEM images of selected nanostar morphology. Lower resolution of −Ag (*a*) and +Ag (*c*) (both with scale bar 50 nm) nanostars. Corresponding nanostars with higher resolution (*b,d*) (both with scale bar 10 nm); (*d*) inset = crystal lattice spacing for +Ag (scale bar 5 nm).

**Figure 5.** Detection of PVP presence. (*a*) NMR spectra obtained for both +Ag and −Ag samples compared with the neat PVP sample. (*b*) Gel electrophoresis showing charge of the particles.

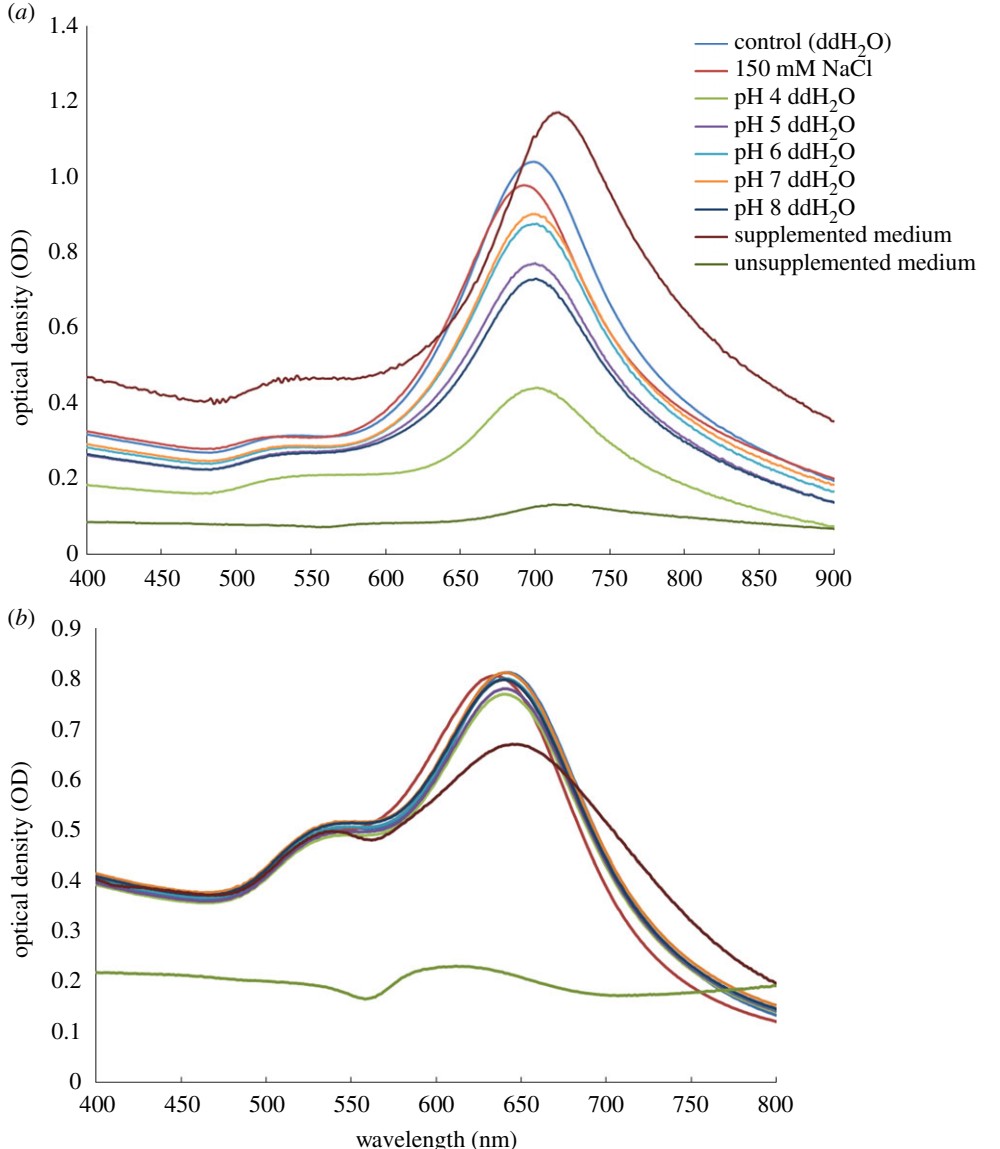

**Figure 6** UV−Vis Spectra of −Ag and +Ag nanostars after a 24 h exposure to various matrices. (*a*) −Ag spectra. (*b*) +Ag spectra. There was a change of the nanostars absorbance over a 24 h time point in various environments, namely a 25% and 17% OD decrease of the −Ag and +Ag in 150 mM NaCl, respectively; a 53% and 6% decrease in pH 4; a 24% and 7% decrease in pH 5; an 18% and 2% decrease in pH 6; a 17% and 3% decrease in pH 7; a 29% and 4% decrease in pH 8; a 33% and 11% decrease in serum-supplemented medium.

phenomenon was a result of highly reactive substrates which were stabilized in the supplemented medium rendering them less reactive with the nanostars [30]. In all remaining instances, the +Ag nanostars were less susceptible to environmental effects as judged by changes in the absorbance and wavelength shifts when compared with the control nanostar solution. In addition, the +Ag nanostars were not significantly affected by changes in the pH 4−8 range. The percentage change in absorbance of both +Ag and −Ag nanostars in the various environments over the 24 hour period are presented in the legend of figure 6.

## 4. Conclusion

We demonstrate better control for more uniform production of gold nanostars morphology. The method showed how a small change in the synthesis method can have a drastic effect on the final morphology of the produced nanostars. This underscores the much needed research in synthesis of gold nanostars and elucidating the mechanism and factors influencing synthesis parameters. We demonstrated how the

addition of silver nitrate significantly enhanced the morphology and monodispersity of a HEPES-mediated seedless nanostar synthesis method. The nanostars resulting from this one-pot synthesis method were approximately $37 \pm 2$ nm in diameter, multi-branched and homogeneous to a large degree. They were also stable in supplemented cell culture medium, sodium chloride and a wide pH range. This green synthesis method is facile and repeatable. It is a safe method for long-term use, thus making it attractive for gold nanostar synthesis for use as scaffolds in biosensors fabrication. Further studies to evaluate this particular feasibility of these nanostars in biosensors are undergoing in our laboratory.

Ethics. Ethical approval to carry out this study was granted by the North-West University Research Ethics Committee.

Data accessibility. Data available from the Dryad Digital Repository: https://doi.org/10.5061/dryad.7nf7d35 [31].

Authors' contributions. D.W.M. conceived the study, designed the study, carried out the synthesis laboratory work, participated in data analysis and drafted the manuscript; A.J. carried out the transmission electron microscopy work and proofread the manuscript; B.C.V. and M.M.P. participated in design of the study, data analysis and drafting of the manuscript. All authors gave final approval for publication.

Competing interests. We declare we have no competing interests.

Funding. All authors were supported by the North-West University's Centre of Human Metabolomics (CHM) and the South African Technology Innovation Agency (TIA) to carry out this work.

Acknowledgements. The authors would like to thank Dr Shayne Mason (https://orcid.org/0000-0002-2945-5768) from the Center for Human Metabolomics, North-West University, Hoffman street, Potchefstroom, South Africa for assistance with obtaining the NMR spectrum.

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
