## [Reviewer comments · Royal Society Open Science]

Review History

RSOS-190160.R0 (Original submission)

Review form: Reviewer 1

Is the manuscript scientifically sound in its present form?

No

Are the interpretations and conclusions justified by the results?

Yes

Is the language acceptable?

Yes

Is it clear how to access all supporting data?

Yes

Do you have any ethical concerns with this paper?

No

Have you any concerns about statistical analyses in this paper?

No

Recommendation?

Accept with minor revision (please list in comments)

Comments to the Author(s)

Vorster and co-workers reported the synthesis of silver-assisted, seedless gold nanostars. The nanostars were monodispersed, multi-branched and stable in various conditions such as salt, ionic strength and cell culture medium environments by functionalized with polyvinylpyrrolidone. In conclusion, the addition of the silver nitrate significantly improves the synthesis of the reported nanostars attaching antibodies and enzymes. In general, the research topic is interesting, and the result is satisfactory, however, there are some issues about experimental details and figures.

The detailed comments are listed as follows:

- 1) Elemental analysis and TEM mapping are recommended for the characterizing gold nanostars.
- 2) Some legend information in Figure 1B are incorrect. The average wavelength at samples of the 8 μ L +Ag should be 598 nm.
- 3) Please note the uniformity of the tenses used in the article.
- 4) For the results of the NMR spectra in Fig. 5A, please mark which group of three curves represents respectively? The resolution of Fig. 5B is low, and the contrast of the figure needs to be adjusted.
- 5) In the caption of Figure 6, "in seum supplemented medium." should be changed to "in serum supplemented medium".
- 6) For environmental effect on nanostar stability, nanostars synthesized by the seeded methods with addition of silver nitrate should also be compared and analyzed.
- 7) The resolution of most figures in the main text is low, and needs improvement.

Review form: Reviewer 2

Is the manuscript scientifically sound in its present form?

No

Are the interpretations and conclusions justified by the results?

No

Is the language acceptable?

Yes

Is it clear how to access all supporting data?

Yes

Do you have any ethical concerns with this paper?

No

Have you any concerns about statistical analyses in this paper?

No

Recommendation?

Reject

Comments to the Author(s)

The manuscript reports on the synthesis of gold nanostars using HEPES as reducing reaction medium and the polymer PVP as stabilizing agent. The authors show that the addition of silver ions improve the quality of the final nanostars in terms of control of number of tips. The presented methodology of gold nanostars synthesis is not novel itself (see for example J. Am. Chem. Soc. 2019, 141, 4034–4042 and its references). Although the paper is interesting and the conclusions are based on the results, I have doubts about the appealing of the study to the broad audience of Royal Society Open Science. The investigation is rather specific because: (i) the purpose of some experiments is not clear (for instance the stabilization at different pHs); (ii) a final objective of this nanoparticles has not been addressed (for example in biomedical application); (iii) there is a lack of presented results (it seems that PVP stabilization has been used only for some type of gold nanostars synthesized in the presence of silver or it has not been analysed the benefits of PVP instead of other polymers such as PEG); and (iv) it is not clear the quality of the final products considering that obtaining better shapes of gold nanostars leads to increments of the formation of spherical gold nanoparticles as byproducts. Therefore, I do question whether Royal Society Open Science is the suitable outlet of this work, and I rather feel that this work will be of interest to a rather specialized subset of authors.

Decision letter (RSOS-190160.R0)

29-Mar-2019

Dear Mrs Mulder:

Title: Modified HEPES One-Pot Synthetic Strategy for Gold Nanostars
Manuscript ID: RSOS-190160

The editor assigned to your manuscript has now received comments from reviewers. We would like you to revise your paper in accordance with the referee and Subject Editor suggestions which can be found below (not including confidential reports to the Editor). Please note this decision does not guarantee eventual acceptance.

Please submit your revised paper before 21-Apr-2019. Please note that the revision deadline will expire at 00.00am on this date. If we do not hear from you within this time then it will be assumed that the paper has been withdrawn. In exceptional circumstances, extensions may be possible if agreed with the Editorial Office in advance. We do not allow multiple rounds of revision so we urge you to make every effort to fully address all of the comments at this stage. If deemed necessary by the Editors, your manuscript will be sent back to one or more of the original reviewers for assessment. If the original reviewers are not available we may invite new reviewers.

On behalf of the Subject Editor Professor Anthony Stace and the Associate Editor Professor Claire Carmalt.

RSC Associate Editor:
Comments to the Author:
(There are no comments.)

RSC Subject Editor:
Comments to the Author:
(There are no comments.)

Reviewers' Comments to Author:
Reviewer: 1

Comments to the Author(s)

Vorster and co-workers reported the synthesis of silver-assisted, seedless gold nanostars. The nanostars were monodispersed, multi-branched and stable in various conditions such as salt, ionic strength and cell culture medium environments by functionalized with polyvinylpyrrolidone. In conclusion, the addition of the silver nitrate significantly improves the synthesis of the reported nanostars attaching antibodies and enzymes. In general, the research topic is interesting, and the result is satisfactory, however, there are some issues about experimental details and figures.

The detailed comments are listed as follows:

- 1) Elemental analysis and TEM mapping are recommended for the characterizing gold nanostars.
- 2) Some legend information in Figure 1B are incorrect. The average wavelength at samples of the 8 μ L +Ag should be 598 nm.
- 3) Please note the uniformity of the tenses used in the article.

- 4) For the results of the NMR spectra in Fig. 5A, please mark which group of three curves represents respectively? The resolution of Fig. 5B is low, and the contrast of the figure needs to be adjusted.
- 5) In the caption of Figure 6, “in seum supplemented medium.” should be changed to “in serum supplemented medium”.
- 6) For environmental effect on nanostar stability, nanostars synthesized by the seeded methods with addition of silver nitrate should also be compared and analyzed.
- 7) The resolution of most figures in the main text is low, and needs improvement.

Reviewer: 2

Comments to the Author(s)

The manuscript reports on the synthesis of gold nanostars using HEPES as reducing reaction medium and the polymer PVP as stabilizing agent. The authors show that the addition of silver ions improve the quality of the final nanostars in terms of control of number of tips. The presented methodology of gold nanostars synthesis is not novel itself (see for example J. Am. Chem. Soc. 2019, 141, 4034–4042 and its references). Although the paper is interesting and the conclusions are based on the results, I have doubts about the appealing of the study to the broad audience of Royal Society Open Science. The investigation is rather specific because: (i) the purpose of some experiments is not clear (for instance the stabilization at different pHs); (ii) a final objective of this nanoparticles has not been addressed (for example in biomedical application); (iii) there is a lack of presented results (it seems that PVP stabilization has been used only for some type of gold nanostars synthesized in the presence of silver or it has not been analysed the benefits of PVP instead of other polymers such as PEG); and (iv) it is not clear the quality of the final products considering that obtaining better shapes of gold nanostars leads to increments of the formation of spherical gold nanoparticles as byproducts. Therefore, I do question whether Royal Society Open Science is the suitable outlet of this work, and I rather feel that this work will be of interest to a rather specialized subset of authors.

Author's Response to Decision Letter for (RSOS-190160.R0)

See Appendix A.

Decision letter (RSOS-190160.R1)

10-May-2019

Dear Mrs Mulder:

Title: Modified HEPES One-Pot Synthetic Strategy for Gold Nanostars
Manuscript ID: RSOS-190160.R1

It is a pleasure to accept your manuscript in its current form for publication in Royal Society Open Science. The chemistry content of Royal Society Open Science is published in collaboration with the Royal Society of Chemistry.

RSC Associate Editor
Comments to the Author:
The revisions are sufficient and the manuscript can be accepted.

Reviewer(s)' Comments to Author:

Appendix A

Response to Reviewers' Comments to Author:

Reviewer 1:

Comments to the Author(s)

Vorster and co-workers reported the synthesis of silver-assisted, seedless gold nanostars. The nanostars were monodispersed, multi-branched and stable in various conditions such as salt, ionic strength and cell culture medium environments by functionalized with polyvinylpyrrolidone. In conclusion, the addition of the silver nitrate significantly improves the synthesis of the reported nanostars attaching antibodies and enzymes. In general, the research topic is interesting, and the result is satisfactory, however, there are some issues about experimental details and figures.

The detailed comments are listed as follows:

1) Elemental analysis and TEM mapping are recommended for the characterizing gold nanostars.

- The elemental analysis was done using EDS and added in the results discussion section. We were however unable to do TEM mapping as this was not available to us.

2) Some legend information in Figure 1B are incorrect. The average wavelength at samples of the 8 μ L +Ag should be 598 nm.

- Thank you for this observation. This has been adjusted accordingly

3) Please note the uniformity of the tenses used in the article.

- Noted. The grammar and tenses were reviewed and corrected throughout the article.

4) For the results of the NMR spectra in Fig. 5A, please mark which group of three curves represents respectively? The resolution of Fig. 5B is low, and the contrast of the figure needs to be adjusted.

- The image was adjusted accordingly and the graphics enhanced.

5) In the caption of Figure 6, "in seum supplemented medium." should be changed to "in serum supplemented medium".

- Noted with thanks. The typing error was corrected.

6) For environmental effect on nanostar stability, nanostars synthesized by the seeded methods with addition of silver nitrate should also be compared and analyzed.

- The seeded method was not compared in this case as this article focused seedless synthesis and the resulting morphology. The suggested comparison would be, however, considered for future study.

7) The resolution of most figures in the main text is low, and needs improvement.

- The figures were adjusted with higher resolutions.

Reviewer: 2

Comments to the Author(s)

The manuscript reports on the synthesis of gold nanostars using HEPES as reducing reaction medium and the polymer PVP as stabilizing agent. The authors show that the addition of silver ions improve the quality of the final nanostars in terms of control of number of tips. The presented methodology of gold nanostars synthesis is not novel itself (see for example J. Am. Chem. Soc. 2019, 141, 4034–4042 and its references).

- Thank you for bringing my attention to the above mentioned article. Silver nitrate has been used extensively in gold nanostars seeded synthesis as a shape directing agent. It has also been used to a much lesser degree in seedless synthesis methods. The use of HEPES has been the predominant seedless shape directing agent. There is, however, room for improvement in the end result of the morphologies obtained by the HEPES synthesis method. The manuscript has been revised to highlight the novel contribution of this study.

Although the paper is interesting and the conclusions are based on the results, I have doubts about the appealing of the study to the broad audience of Royal Society Open Science. The investigation is rather specific because:

- (i) The purpose of some experiments is not clear (for instance the stabilization at different pHs);
- This was stated in the manuscript, however, has been rewritten to make it clearer. It now reads as “These parameters were chosen as most bioassay work would require the nanostars to be stable in these conditions.”

(ii) A final objective of this nanoparticles has not been addressed (for example in biomedical application);

- The concluding statement has been modified to clarify this and now reads as “This green synthesis method is facile and repeatable. It is a safe method for long term use, thus making it attractive for gold nanostar synthesis for use as scaffolds in biosensors fabrication. Further studies to evaluate this particular feasibility of these nanostars in biosensors are undergoing in our laboratory.”

(iii) There is a lack of presented results (it seems that PVP stabilization has been used only for some type of gold nanostars synthesized in the presence of silver or it has not been analysed the benefits of PVP instead of other polymers such as PEG).

- An adjustment was made to make this point clearer. It now reads as “The stabilization of other coating polymers was not assessed as this was done in another study found in literature using HEPES buffer. What was found was that PEG was not a good stabilising agent compared to PVP and CTAB (1). PVP was, thus, chosen as it was the less toxic reagent when compared to CTAB (2, 3).”

(iv) It is not clear the quality of the final products considering that obtaining better shapes of gold nanostars leads to increments of the formation of spherical gold nanoparticles as byproducts. Therefore, I do question whether Royal Society Open Science is the suitable outlet of this work, and I rather feel that this work will be of interest to a rather specialized subset of authors.

- Each synthesis method has its own yield rate where other morphologies are by products. Not all synthesis methods yield rates are documented in literature, however, here is an article which did show their yield rate (4). This information was added to the manuscript. “Liu et. al synthesized nanostars with four long branched arms by manipulating the HEPES concentration and solution temperature. The yield rates obtained for the nanostars were 65% tetrapods, 6% five branches or more, 15% two-three branches and 14% spherical and irregular shapes (4).”

Royal Society Open Science Journal covers the entire range of science without the usual restrictions on scope. We, therefore, believe our manuscript is suitable for publication by as the journal is open to many readers of different disciplines, such as nanotechnology.

1. Chen R, Wu J, Li H, Cheng G, Lu Z, Che C-M. Fabrication of gold nanoparticles with different morphologies in HEPES buffer. *Rare Metals*. 2010;29(2):180-6.
2. Alkilany A, Murphy C. Toxicity and cellular uptake of gold nanoparticles: What we have learned so far? 2010. 2313-33 p.
3. Koczkur KM, Mourdikoudis S, Polavarapu L, Skrabalak SE. Polyvinylpyrrolidone (PVP) in nanoparticle synthesis. *Dalton Transactions*. 2015;44(41):17883-905.
4. Liu H, Xu Y, Qin Y, Sanderson W, Crowley D, Turner CH, et al. Ligand-Directed Formation of Gold Tetrapod Nanostructures. *The Journal of Physical Chemistry C*. 2013;117(33):17143-50.